# Research on the Fire Extinguishing Efficiency of Low-Pressure Water Mist in Urban Underground Utility Tunnel Cable Fires

Boyan Jia [1], Yanwei Xia [1], Zhaoyu Ning [2,*], Bin Li [2], Guowei Zhang [2,3] and Zhiwei Zhang [3]

1   State Grid Hebei Electric Power Research Institute, Shijiazhuang 050011, China
2   School of Safety Engineering, China University of Mining and Technology, Xuzhou 221116, China
3   Shenzhen Research Institute, China University of Mining and Technology, Shenzhen 518000, China
*   Correspondence: nzy@cumt.edu.cn

**Abstract:** Low-pressure water mist fire extinguishing systems are a cost-effective and highly reliable option for fire protection. However, they have not yet seen widespread use in urban underground utility tunnels. To validate the fire extinguishing effectiveness of the system in cable fires within urban utility tunnels and to identify the key factors influencing its efficiency, a scaled-down test platform for low-pressure water mist fire extinguishing in utility tunnels was constructed, and a series of fire extinguishing tests was conducted. The test results demonstrate that low-pressure water mist can rapidly and effectively extinguish cable fires in utility tunnels, with the quickest fire extinguishing time of 7 s. Within 50 s of activating the system, the internal temperature of the tunnel can be reduced from 650 °C to 40 °C. Among the influencing factors, the pressure and nozzle flow coefficient have a significant impact on the fire extinguishing efficiency, while nozzle spacing has a relatively smaller effect. Thus, when the nozzle spacing meets the requirement of "no dead zones", priority should be given to increasing the pressure and nozzle flow coefficient.

**Keywords:** low-pressure water mist; utility tunnel; cable fire; fire extinguishing efficiency

## 1. Introduction

Urban underground utility tunnels are one of the key construction projects in China, playing a crucial role in supporting the normal operation of cities [1]. The utility tunnel not only saves urban space but also facilitates the management and maintenance of pipelines [2,3]. With strong policy support, the total mileage of urban underground utility tunnels in China has rapidly increased, making it the world's leader [4]. However, the frequent occurrence of fire accidents in these tunnels has drawn widespread public attention. The cabins of urban underground utility tunnels are filled with various pipelines such as cables, optical fibers, water supply pipes, and water drainage pipes, posing a high fire load and presenting significant challenges to fire extinguishing and rescue efforts [5]. Among these, cables pose the highest fire risk and are the main cause of fires in utility tunnels [6].

Currently, water mist fire extinguishing systems are primarily used in cable cabins of urban underground utility tunnels. The water mist fire extinguishing system is a new, efficient, and environmentally friendly fire extinguishing system. Scholars have conducted a series of research on water mist fire extinguishing systems in utility tunnels. Li et al. [7] conducted numerical simulations of water mist fire extinguishing in utility tunnels by changing parameters such as the position of the fire source, the number of nozzles, average droplet size, and nozzle pressure. They found that the high-pressure water mist fire extinguishing system has a good cooling effect, and the droplet size has a major impact on fire extinguishing efficiency. Huang et al. [8] carried numerical simulations by the Fire Dynamics Simulator (FDS) to analyze the variations in nozzle activation time, fire suppression time, and ignited cable length under different conditions. They found that nozzle activation time should decrease with increasing nozzle spacing. Xu et al. [9] conducted experimental

studies on the full submersion and local application of water mist fire extinguishing systems under different conditions. They found that for electrical cabins in utility tunnels, a full submersion water mist fire extinguishing system is preferable. M Pokorný et al. [10] conducted large-scale fire tests in a room with dimensions of $4.0 \times 4.0 \times 2.0$ m and found that low-pressure water mist effectively protects steel structures from fire. Wu et al. [11] studied the fire extinguishing behavior of cable tunnel fires using scaled-down experiments and found that the fire extinguishing effectiveness of water mist is influenced by ventilation conditions and the arrangement and quantity of combustibles. Chen et al. [12] discussed the fire extinguishing effectiveness for cable cabin fires using water mist with different particle sizes through physical experiments combined with FDS simulations. They found that water mist with a particle size of 50–100 μm is effective for extinguishing cable cabin fires. Chai et al. [13] studied different fire extinguishing systems in full-scale tests in utility tunnels and found that high-pressure water mist has better cooling and reignition prevention effects. Roberto et al. [14] used FDS to simulate the low-pressure water mist fire extinguishing behavior for ship engine compartment fires, showing good consistency with real experiments in terms of compartment temperature change trends and fire extinguishing time. Zhao et al. [15] successfully suppressed thermal runaway in a lithium-ion battery box using low-pressure water mist in experiments.

The research of these scholars has validated the effectiveness of water mist fire extinguishing systems, but the studies are mostly based on high-pressure systems, while there is almost no research on low-pressure water mist fire extinguishing in cable fires in utility tunnels. According to the National Fire Protection Association [16], a low-pressure water mist system is defined as a water mist system where the distribution piping is exposed to pressures of 175 psi (12.1 bars) or less. Although the main difference between high-pressure and low-pressure water mist is pressure, the difference in pressure leads to variations in system design, nozzle types, and water droplet sizes, all of which can impact the system's performance. Furthermore, due to the high pressure requirements of high-pressure water mist fire extinguishing systems, which demand high standards for pumps and pipelines, the required civil construction costs and subsequent maintenance expenses are high. In contrast, low-pressure water mist fire extinguishing systems have lower pressure requirements for power supply, pipes, fittings, and valves, making them more reliable and cost-effective. If widely applied in utility tunnels, they could significantly reduce the costs of safety input during the construction and operation of utility tunnels, promoting the rapid development of utility tunnels in the world.

In summary, scaled-down tests were conducted, verifying the effectiveness of low-pressure water mist fire extinguishing systems in urban underground utility tunnel cable fires. Furthermore, the impact of different nozzle flow coefficients, nozzle spacings, and pressures on the fire extinguishing effectiveness of low-pressure water mist was analyzed, identifying key parameters affecting its efficiency. The results of this research can provide a theoretical foundation and data support for the engineering application of low-pressure water mist fire extinguishing systems in urban underground utility tunnels.

## 2. Test Design

### 2.1. Test System and Devices

To accurately replicate the real conditions of cable fires in urban underground utility tunnels and investigate the extinguishing characteristics of low-pressure water mist, a scaled-down test platform was constructed, as shown in Figure 1. The utility tunnel model (10.0 m long, 0.9 m wide in cross-section, and 1.25 m high) is scaled down from a real tunnel of 2.7 m width and 3.75 m height; the dimensionless scale ratio is thus S = 1/3. The similarity theory is based on geometric and dynamic similarity to ensure that the results obtained from these tests can reveal the relevant characteristics of the real conditions of cable fires in urban underground utility tunnels. The model is made mainly of high-temperature-resistant stainless steel, and the materials for the front and both sides of the model are made of high-temperature-resistant quartz glass, allowing real-time observation of the

combustion phenomena, fire development, and low-pressure water mist fire extinguishing during the testing process.

The low-pressure water mist fire extinguishing system consists of water mist pipelines, zone valves, a lightweight centrifugal pump, and a water storage tank. The rated pressure of the pump is 1.6 MPa. A return-flow pipeline is located on the left side of the pump, and the pressure within the pipeline is controlled by adjusting the valve on the return-flow pipeline. The water storage tank is made of PVC material with a volume of 1.5 m³. The flow rate of water mist and the water storage capacity of the tank can be calculated using the following formulae [17].

$$q = K \sqrt{10\,p} \qquad (1)$$

$$Q_s = \sum_{i=1}^{n} q_i \qquad (2)$$

$$V = Q_s \cdot t \qquad (3)$$

Among them, $q$ is the design flow rate of the nozzle, L/min; $K$ is the flow coefficient of the nozzle, nondimensional; $p$ is the design working pressure of the nozzle, MPa; $Q_s$ is the design flow rate of the system, L/min; $n$ is the number of nozzles; $V$ is the required effective volume of the water storage tank, L; and $t$ is the design spray time of the system, min.

Upon powering on the pump, water is drawn from the storage tank and flows into the utility tunnel model through pipelines. The zone valve switch is used to control the activation of the low-pressure water mist fire extinguishing system. There are three nozzles at the top of the utility tunnel model, as shown in Figure 1a,b. Among these, the central nozzle remains fixed, and adjustments in the nozzle spacing only affect the distance between the other two nozzles and the central one.

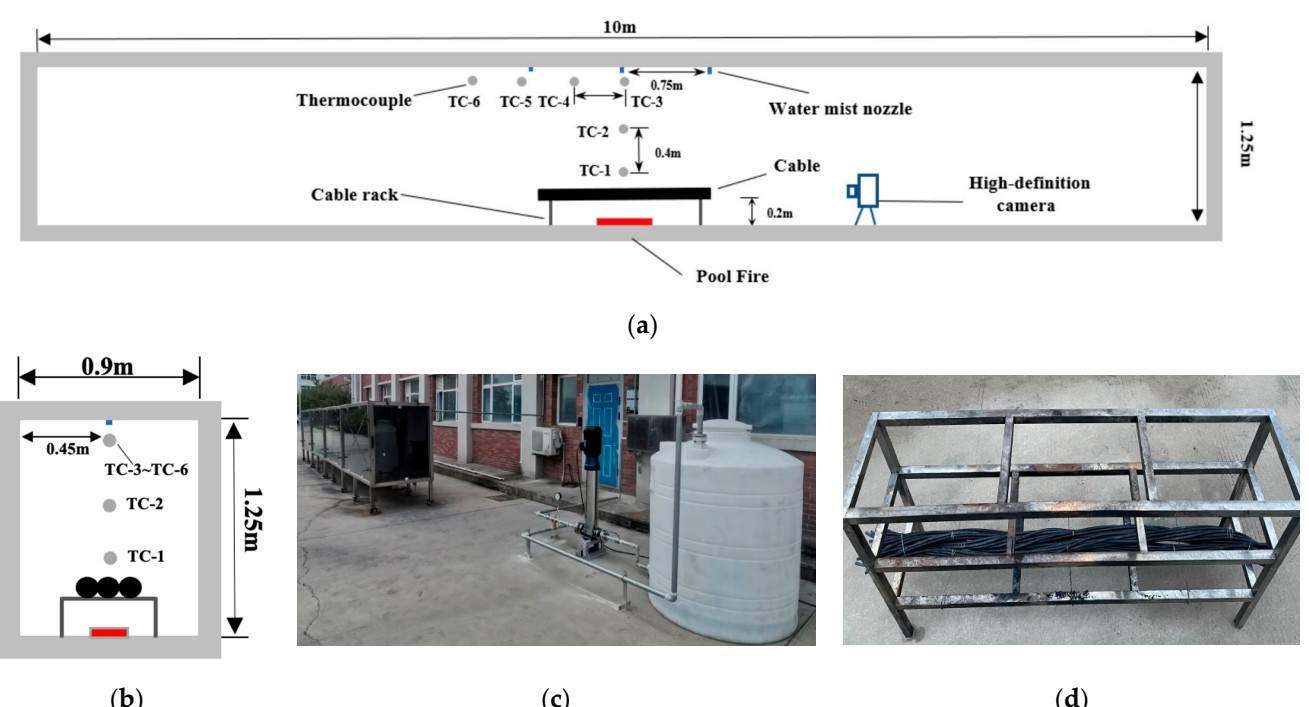

**Figure 1.** Low-pressure water mist fire extinguishing test platform for the scaled-down utility tunnel: (**a**) front view of the system setup; (**b**) side view of the system setup; (**c**) photo of the test site; (**d**) photo of the cables and the cable rack.

The cable rack is constructed of stainless steel and placed close to the model's sidewall, as shown in Figure 1d. The cables used are TVR general-purpose flexible cables, each 1.5 m long. A total of 30 cables is securely arranged and placed on the cable rack, positioned at a

height of 0.2 m above the model ground. The ignition source consists of a rectangular oil pool measuring 0.6 m × 0.15 m × 0.02 m, located directly beneath the cable rack. N-heptane is used as the fuel, which burns steadily and produces minimal black smoke, facilitating observation during the experiments.

The primary mechanisms of low-pressure water mist fire extinguishing are cooling and asphyxiation [18]. In the case of a utility tunnel cable fire, low-pressure water mist directly affects the flames, resulting in a pronounced cooling effect. Therefore, temperature data are taken and analyzed in detail. Six K-type thermocouples are arranged longitudinally and laterally within the utility tunnel to measure temperature data during the tests. Longitudinal thermocouples are placed directly above the cables, neatly spaced along the centerline of the cable rack, with a 0.4 m gap between each thermocouple. The lower row of thermocouples is positioned 0.4 m above the ground, while the upper row of thermocouples is placed 1.2 m above the ground. At a height of 1.2 m above the ground, lateral thermocouples are placed at 0.4 m intervals, with the outermost thermocouple located 1.2 m from the centerline of the cable rack. The serial number of thermocouples is marked as TC-1 to TC-6, as shown in Figure 1a,b.

### 2.2. Test Conditions Setup

The flow coefficient of the nozzle *K*, pressure *p*, and nozzle spacing *d* are the three main parameters that affect the fire extinguishing effectiveness of low-pressure water mist. To validate the fire extinguishing effectiveness of low-pressure water mist for cable fires in the utility tunnel, the test condition with *K* of 1.6, *P* of 1 MPa, and *d* of 750 mm was selected for the test, marked as Test 1.

Furthermore, to investigate the impact of *K*, *p*, and *d* on the fire extinguishing effectiveness of low-pressure water mist for cable fires in the utility tunnel, tests were conducted with these three parameters as variables. Test 1 was selected as the control group, and the whole test conditions are outlined in Table 1.

**Table 1.** Test conditions.

| Test Number | *K* | *p* (MPa) | *d* (mm) | Remarks |
|:---:|:---:|:---:|:---:|:---:|
| 1 | 1.6 | 1 | 750 | Control group |
| 2 | 0.5 | 1 | 750 | |
| 3 | 0.8 | 1 | 750 | Change flow coefficient of the nozzle |
| 4 | 2.3 | 1 | 750 | |
| 5 | 1.6 | 1 | 500 | |
| 6 | 1.6 | 1 | 900 | Change nozzle spacing |
| 7 | 1.6 | 1 | 1250 | |
| 8 | 1.6 | 0.6 | 750 | |
| 9 | 1.6 | 0.8 | 750 | Change pressure |
| 10 | 1.6 | 1.2 | 750 | |

### 2.3. Test Method

The test officially commenced by igniting the oil pool with an extended igniter. The oil pool continued to burn and ignited the cables. At 150 s, the cable fire entered a stable burning phase, at which point the low-pressure water mist fire extinguishing system was activated. The low-pressure water mist fire extinguishing system remained active for 360 s, continuing to cool the utility tunnel model even after the flames were extinguished. After 360 s of operation, the interior temperature of the utility tunnel model returned to ambient temperature, marking the end of the test.

## 3. Analysis of Utility Tunnel Low-Pressure Water Mist Fire Extinguishing Test Results

*3.1. Validation of Fire Extinguishing Effectiveness*

Figure 2 illustrates the test phenomenon of Test 1. At 0 s, the test officially commenced by igniting the oil pool. As the combustion progressed, the flames from the oil pool gradually expanded and ignited the cables at 30 s. By 90 s, the cable fire continued to intensify. At 150 s, the low-pressure water mist fire extinguishing system was opened, initiating the fire extinguishing process.

Upon opening the zone valve switch, water within the pipelines rapidly transformed into water mist and was sprayed into the utility tunnel model, creating strong turbulence around the fire source. At 151 s, the flames were suppressed immediately by the impact of the water mist, resulting in a rapid reduction in flame height. By 158 s, under the continuous suppression of the water mist, the flames exhibited noticeable unstable fluctuations. At 164 s, the flames were completely extinguished. The middle section of the cables presented a charred appearance after the fire was extinguished, as shown in Figure 3.

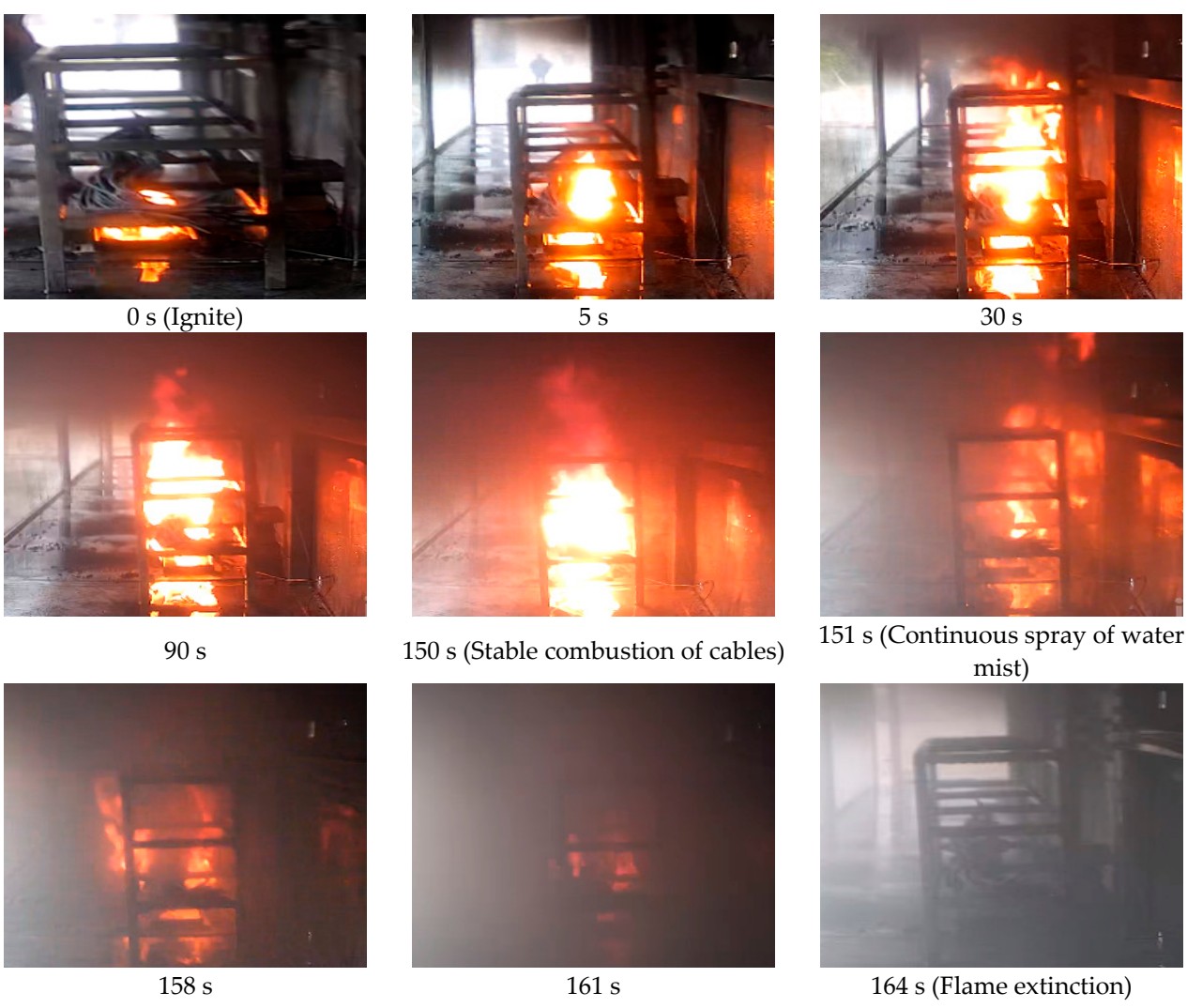

**Figure 2.** Test phenomenon of Test 1.

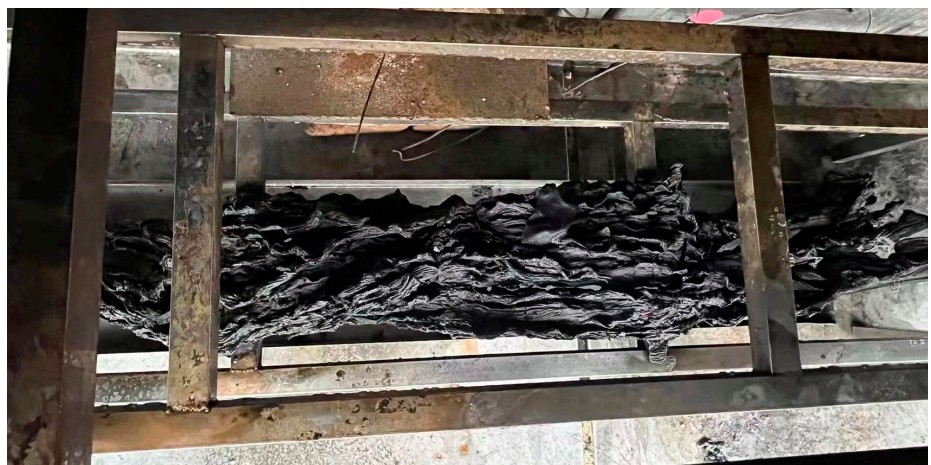

**Figure 3.** Photo of the charred cables.

Figure 4 illustrates the temperature changes in the thermocouples under Test 1. At a height of 0.4 m (TC-1), the measured temperature corresponds to the flame temperature. After the ignition, the temperature of TC-1 exhibits a rapid increase and reaches approximately 770 °C at around 40 s. At around 150 s, after reaching the fully developed phase, the temperature stabilizes and fluctuates around 600 °C, with a deviation of approximately 50 °C. At the ceiling directly above the fire source (TC-3), the temperature continues to rise due to the continuous impact of flames and the heat exchange with high-temperature smoke. At around 150 s, it reaches the highest temperature of approximately 400 °C. On the left side of the ceiling (TC-4, TC-5, and TC-6), the temperature keeps steadily and gradually increases after ignition, without fluctuations. This trend differs from the temperature changes in TC-1, TC-2, and TC-3 because the left side of the ceiling is not directly heated by flames, resulting in less influence from flame fluctuations.

At 150 s, the injection of low-pressure water mist has a pronounced cooling effect on both the flames and the ceiling. Specifically, in the flame area (TC-1, TC-2), the temperature exhibits a significant decreasing trend in the initial stages following the activation of the fire extinguishing system. After 18 s of system activation, the temperature has already decreased by 50%, with visible flames extinguished at this point. After 50 s of system activation, the internal tunnel temperature drops from the highest temperature, 650 °C, to 40 °C.

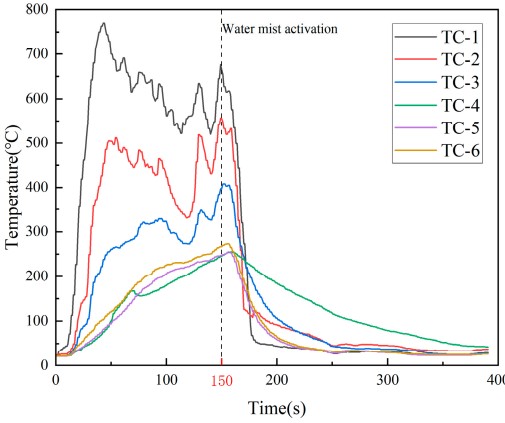

**Figure 4.** Temperature change curve of each thermocouple.

At the ceiling directly above the flame (TC-3), the temperature decreases rapidly in the initial stages of system activation, while exhibiting a slower decline between 35 s and 70 s of system activation. This is because the droplet size of the fine water mist is small, the specific surface area is large, and the surface heat transfer coefficient is large. When

the fire extinguishing system is initially activated, the ambient temperature inside the model is high. Upon system activation, water mist droplets vaporize rapidly, absorbing a significant amount of ambient heat. The dense water mist droplets effectively block the intensity of heat radiation from flames and the convection heat from smoke, leading to a rapid reduction in the fire temperature. Furthermore, some high-velocity mist droplets impact the surfaces of cables, thereby facilitating the wetting of the combustible materials, ultimately achieving the objective of fire suppression and preventing the fire from spreading. As the fire extinguishing system continues to operate, the temperature difference between the hot smoke layer and the water mist gradually decreases, resulting in a reduced rate of heat absorption through water mist evaporation.

On the left side of the ceiling (TC-4, TC-5, and TC-6), the temperature continues to increase in the first 10 s of system activation. Moreover, the rate of temperature decrease is significantly slower than the other three thermocouples. This indicates that water mist mainly cools the flame area within its coverage range, while its cooling effect on the surrounding environment is limited, resulting in a lag in temperature changes in areas far from the flames compared to those in the flame area.

In conclusion, the low-pressure water mist fire extinguishing system can effectively extinguish utility tunnel cable fires. The cooling effect of the low-pressure water mist fire extinguishing system is highly noticeable, and the fire extinguishing time is short.

### 3.2. Analysis of the Impact of the Flow Coefficient of the Nozzle on Low-Pressure Water Mist Fire Extinguishing Efficiency

As shown in Figures 3 and 4, due to the direct impact of flames, the temperature fluctuation range of TC-1, TC-2, and TC-3 is measurable, which affects the investigation of the cooling effect of the low-pressure water mist fire extinguishing system. The left-side area of the ceiling (TC-4, TC-5, and TC-6) is not directly impacted by the flames, resulting in less influence from flame fluctuations. Therefore, the temperature data from TC-6 is selected for analysis.

Table 2 presents the results of four sets of test conditions under different nozzle flow coefficients ($k$). The table shows that as $K$ increases, the fire extinguishing time significantly shortens. The fire extinguishing time for Test 4 is 10 s, which is a 66.7% reduction compared to the fire extinguishing time (30 s) for Test 2. The fire extinguishing time for Test 1 and Test 4 is close. This indicates that if $K$ continues to increase, the fire extinguishing time will not see a substantial further reduction.

Figure 5 illustrates the temperature variations at TC-6 under different nozzle flow coefficients. In Figure 5b, each vertical line indicates the moment when the temperature in each condition drops to half of the maximum temperature. The trend of temperature reduction is smoother for Test 2 and Test 3 after initiating the low-pressure water mist fire extinguishing system. Test 1 and Test 4 show a rapid decrease in temperature, and their temperature reduction trends are consistent. At the same time, the moments when their temperatures reach half of the maximum temperature are close. This indicates that as $K$ increases from 1.6 to 2.3, the cooling effect does not increase significantly. This is because an increase in $K$ results in higher water usage, leading to an increased number of mist droplets reaching the fire area. Within a certain effective area, the ejected water mist absorbs more heat from the surrounding fire scene, leading to a faster temperature reduction and, consequently, improved fire suppression efficiency. However, when $K$ reaches a certain point, the ejected mist droplets tend to saturate, preventing further effective enhancement of the cooling effect.

**Table 2.** Results of 4 sets of tests with different nozzle coefficients (temperature data for TC-6).

| Test | $K$ | Highest Temperature (°C) | Temperature at Time of Fire Extinction (°C) | Fire Extinguishing Time (s) | Average Pre-Extinguishing Cooling Rate (°C/s) | Time When Temperature Drops to Half of the Maximum (s) |
|---|---|---|---|---|---|---|
| 2 | 0.5 | 226 | 139 | 30 | 2.9 | 45 |
| 3 | 0.8 | 240 | 140 | 26 | 3.85 | 33 |
| 1 | 1.6 | 273 | 195 | 14 | 5.57 | 29 |
| 4 | 2.3 | 252 | 170 | 10 | 8.2 | 28 |

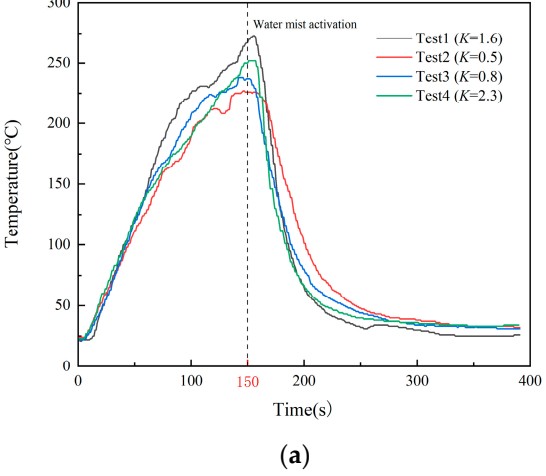 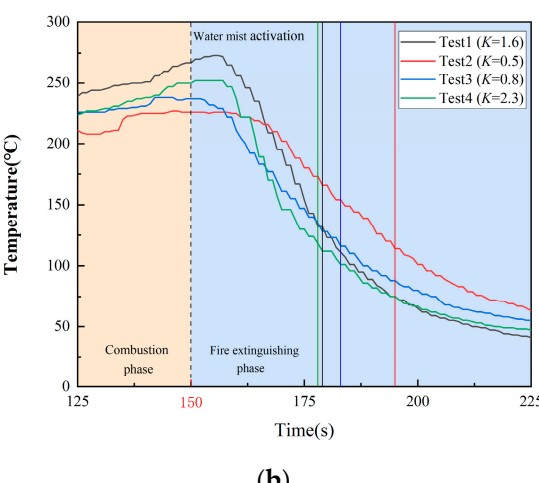

(a)     (b)

**Figure 5.** Temperature change curves of TC-6 under different nozzle flow coefficients: (**a**) full test duration; (**b**) partial enlargement.

In summary, it is evident that when $K$ is low, increasing it within a certain range significantly enhances the cooling effect and fire extinguishing efficiency of the low-pressure water mist fire extinguishing system, reducing the fire extinguishing time. However, when $K$ continues to increase from 1.6 to 2.3, the cooling effect of low-pressure water mist does not show a significant improvement.

### 3.3. Analysis of the Impact of Nozzle Spacing on Low-Pressure Water Mist Fire Extinguishing Efficiency

Table 3 provides the results of four sets of tests under different nozzle spacings ($d$). From Table 3, it can be observed that with an increasing $d$, the variation in fire extinguishing time is relatively small. Both Test 1 and Test 6 have a fire extinguishing time of 14 s. Test 5 has a fire extinguishing time of 10 s, which is 47.3% lower than Test 7 (19 s).

Figure 6 presents the temperature changes at TC-6 under different nozzle spacings. The four test results exhibit similar patterns in terms of peak temperature, temperature at the time of fire extinction, and the average pre-extinguishing cooling rate. Test 5 shows only a slight advantage in cooling effectiveness. It is worth noting that Test 7 has a higher pre-extinguishing average cooling rate compared to Test 1 and Test 6, and the time when the temperature drops to half of the maximum is shorter. This is because decreasing $d$ for water mist means a greater number of nozzles, resulting in more water mist filling the entire utility tunnel. As a result, the cooling effect is enhanced, and the fire suppression time is reduced. However, when $d$ increases, the coverage area (or protection radius) of water mist expands. Consequently, the fire extinguishing effectiveness in the overlapping regions of different nozzles decreases. Test 5 has the smallest $d$, resulting in a more concentrated effect of nozzles on the fire source, The flames extinguish earlier, resulting in a decrease in ambient temperature. In contrast, the larger $d$ in Test 7 significantly increases the nozzle's

coverage area, providing better cooling effects in the surroundings of the fire source before extinguishing it.

**Table 3.** Results of 4 sets of tests with different nozzle spacings (Temperature data for TC-6).

| Test | $d$ (mm) | Highest Temperature (°C) | Temperature at Time of Fire Extinction (°C) | Fire Extinguishing Time (s) | Average Pre-Extinguishing Cooling Rate (°C/s) | Time When Temperature Drops to Half of the Maximum (s) |
|------|-----|-----|-----|-----|------|-----|
| 5 | 500 | 257 | 196 | 10 | 6.1 | 22 |
| 1 | 750 | 273 | 195 | 14 | 5.57 | 29 |
| 6 | 900 | 269 | 198 | 14 | 5.07 | 32 |
| 7 | 1250 | 265 | 157 | 19 | 5.68 | 25 |

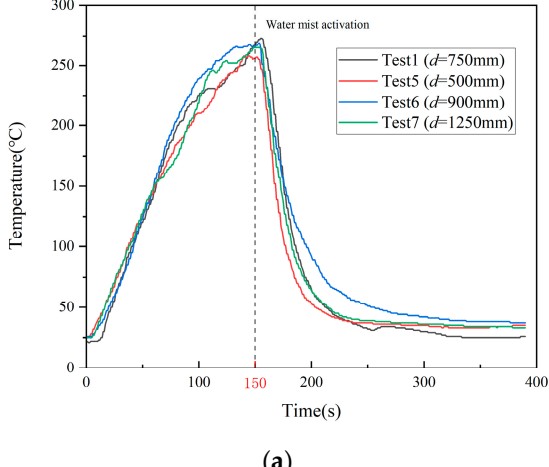
(**a**)

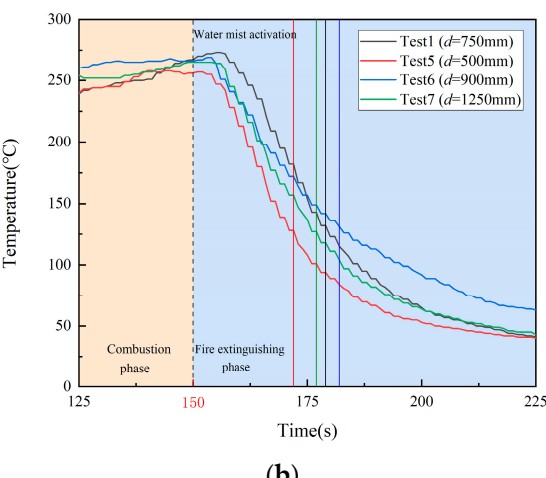
(**b**)

**Figure 6.** Temperature change curves of TC-6 under different nozzle spacings: (**a**) full test duration; (**b**) partial enlargement.

In conclusion, changes in $d$ have a certain degree of impact on the cooling efficiency of the low-pressure water mist fire extinguishing system. A smaller d achieves better fire extinguishing results, while a larger $d$ results in a larger coverage area for low-pressure water mist. Taking into account both fire extinguishing efficiency and the impact on the protection radius, the condition with a $d$ of 750 mm yields the optimal results.

### 3.4. Analysis of the Impact of Pressure on Low-Pressure Water Mist Fire Extinguishing Efficiency

Table 4 presents the results of four different tests under different pressures ($p$). From Table 4, it is evident that with an increase in $p$, the fire extinguishing time significantly decreases. In Test 10, the fire extinguishing time is 7 s, which is a 75.0% reduction compared to Test 8 (28 s) and a 50% reduction compared to Test 1 (14 s). This indicates that $p$ has a substantial impact on the fire extinguishing effectiveness of low-pressure water mist.

Figure 7 presents the temperature changes at TC-6 under different pressures. From Figure 7, it can be observed that there are significant differences in the temperature reduction trends. In particular, under Test 8, which has the lowest $p$, the temperature reduction trend is the most gradual. As $p$ increases, the ejected mist droplets have smaller particle sizes, resulting in a larger specific surface area of the water mist. This allows it to absorb more heat within the surrounding space, leading to a significantly enhanced rate of temperature reduction. The average pre-extinguishing cooling rate for Test 10 is 7.43 °C/s, which is 2.6 times that of Test 8 (2.89 °C/s).

**Table 4.** Results of 4 sets of tests with different pressures (temperature data for TC-6).

| Test | $p$ (MPa) | Highest Temperature (°C) | Temperature at Time of Fire Extinction (°C) | Fire Extinguishing Time (s) | Average Pre-Extinguishing Cooling Rate (°C/s) | Time When Temperature Drops to Half of the Maximum (s) |
|------|-----------|--------------------------|---------------------------------------------|-----------------------------|-----------------------------------------------|--------------------------------------------------------|
| 8 | 0.6 | 248 | 167 | 28 | 2.89 | 46 |
| 9 | 0.8 | 271 | 205 | 18 | 3.67 | 34 |
| 1 | 1.0 | 273 | 195 | 14 | 5.57 | 29 |
| 10 | 1.2 | 245 | 193 | 7 | 7.43 | 25 |

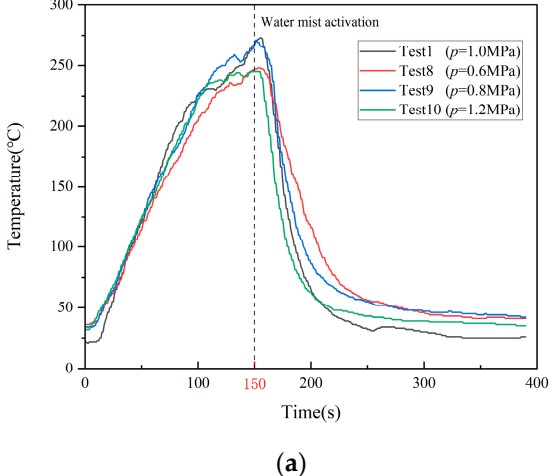

(**a**)

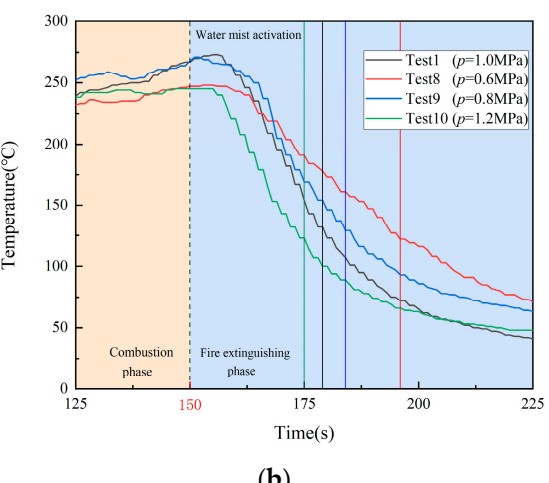

(**b**)

**Figure 7.** Temperature change curves of TC-6 under different spacings: (**a**) full test duration; (**b**) partial enlargement.

In summary, changes in *p* have a significant impact on the fire extinguishing efficiency of the low-pressure water mist fire extinguishing system. As *p* increases, the cooling effect and fire extinguishing efficiency of the low-pressure water mist fire extinguishing system significantly improve.

## 4. Conclusions

In order to investigate the fire extinguishing behavior and efficiency of the low-pressure water mist fire extinguishing system in urban utility tunnel cable fires, a scaled-down test platform was constructed. A series of fire extinguishing tests was conducted with the nozzle flow coefficient, nozzle spacing, and pressure as variables. A quantitative comparative analysis of these three factors influencing the fire extinguishing efficiency of the low-pressure water mist fire extinguishing system was performed. The following main conclusions were obtained:

(1) The low-pressure water mist fire extinguishing system is effective in extinguishing cable fires in utility tunnels. It can effectively extinguish flames in tests of all conditions, with the shortest fire extinguishing time of 7 s. The cooling effect of the low-pressure water mist fire extinguishing system is outstanding, and the system can reduce the temperature inside the tunnel from 650 °C to 40 °C within 50 s of system activation.

(2) The pressure and nozzle flow coefficient significantly affect the fire suppression efficiency of the low-pressure water mist fire extinguishing system, while nozzle spacing has a smaller impact. Changes in pressure, nozzle flow coefficient, and nozzle spacing can, respectively, reduce the fire extinguishing time by a maximum of 75%, 66.7%, and 47.3%.

(3) As inferred from the previous conclusion (2), reducing nozzle spacing enhances the effectiveness of low-pressure water mist fire suppression in utility tunnels. However,

the improvement in effectiveness is relatively modest compared to adjustments in nozzle coefficients and increased pressure. Furthermore, in practical engineering applications, reducing nozzle spacing leads to an increased number of nozzles, resulting in higher deployment and maintenance costs. Therefore, for low-pressure water mist systems, when the nozzle spacing of the low-pressure water mist fire extinguishing system meets the requirement of "no dead zones", priority should be given to increasing the system's pressure and the flow coefficient of the nozzles.

**Author Contributions:** Conceptualization, B.J. and Y.X.; methodology, B.J.; software, B.J.; validation, B.J., Y.X. and G.Z.; formal analysis, B.J. and Z.N.; investigation, Z.N. and B.L.; resources, Z.Z.; data curation, Z.N.; writing—original draft preparation, B.J. and Y.X.; writing—review and editing, Z.N. and Z.Z.; visualization, B.L. and Z.Z.; supervision, Y.X.; project administration, G.Z.; funding acquisition, B.J. All authors have read and agreed to the published version of the manuscript.

**Funding:** This research was funded by the State Grid Hebei Electric Power Co., LTD., grant number kj2022-061; and the National Fire and Rescue Administration, grant number 2022XFZD01.

**Institutional Review Board Statement:** Not applicable.

**Informed Consent Statement:** Informed consent was obtained from all subjects involved in the study.

**Data Availability Statement:** No data were used to support this study. However, any query about the research conducted in this paper is highly appreciated and can be asked from the corresponding authors upon request.

**Conflicts of Interest:** The authors declare no conflict of interest.

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
