# Peer review of "Research on the Fire Extinguishing Efficiency of Low-Pressure Water Mist in Urban Underground Utility Tunnel Cable Fires"

_fire, doi:10.3390/fire6110433_

Round 1
Reviewer 1 Report
Comments and Suggestions for Authors
I would like to congratulate you on your beautifully presented research paper. There are a few suggestions/comments that could improve your work.
1. Please prepare the manuscript according to the journal template.
2. It would be nice if you refer to your similar work https://doi.org/10.3390/fire5060202
3. Does the scaling have any impact and what is the scaling factor? Please address this question.
4. What influence do the glass sides have on the temperature development compared to the real walls of the tunnel?
5. Pg. 3 - Please delete the Chinese letters above equations (1)-(3).
6. Fig. 3 - Zoom on cable would be better for the illustration.
7. In general, the diagrams should correspond to the template... make them more readable (enlarge font...).
8. Conclusion (3) should be reconsidered. Delete 'Therefore,' and please be more specific... because this statement (3) is too general.
Comments on the Quality of English Language
Minor editing of English language required.
Reviewer 2 Report
Comments and Suggestions for Authors
In this work, the effectiveness of low-pressure water mist fire extinguishing systems in urban underground utility tunnel cable fire was analyzed by using small-scale experiments. Analyzed the effects of different nozzle flow coefficients, nozzle spacing, and pressure, and determined the key parameters that affect their firefighting effectiveness.
The theme of this paper is interesting, and my main comments are as follows:
1. What similarity theory is used for the research conducted through small-scale experiments in the paper? What is the scale of the model? Please provide additional information on model similarity.
2. This article mainly introduces the experimental results, which is more like an experimental report and lacks theoretical research and deeper analysis. Suggest further analysis of the internal mechanism of the fire extinguishing effect of fine water mist.
3. The author mainly studies the fire extinguishing effectiveness of low-pressure water mist, and what is the definition of low-pressure water mist. Is it just different from high-pressure water mist in terms of pressure?
Round 2
Reviewer 2 Report
Comments and Suggestions for Authors
I don't have other comments